# Effects of Pitavastatin, Atorvastatin, and Rosuvastatin on the Risk of New-Onset Diabetes Mellitus: A Single-Center Cohort Study

**DOI:** 10.3390/biomedicines8110499

**Published:** 2020-11-13

**Authors:** Wei-Ting Liu, Chin Lin, Min-Chien Tsai, Cheng-Chung Cheng, Sy-Jou Chen, Jun-Ting Liou, Wei-Shiang Lin, Shu-Meng Cheng, Chin-Sheng Lin, Tien-Ping Tsao

**Affiliations:** 1Department of Internal Medicine, Tri-Service General Hospital, National Defense Medical Center, Taipei 11490, Taiwan; joe800115@hotmail.com; 2School of Public Health, National Defense Medical Center, Taipei 11490, Taiwan; xup6fup0629@gmail.com; 3School of Medicine, National Defense Medical Center, Taipei 11490, Taiwan; 4Graduate Institute of Life Sciences, National Defense Medical Center, Taipei 11490, Taiwan,; 5Department of Physiology and Biophysics, Graduate Institute of Physiology, National Defense Medical Center, Taipei 11490, Taiwan; mctsaisyy@gmail.com; 6Division of Cardiology, Department of Internal Medicine, Tri-Service General Hospital, National Defense Medical Center, Taipei 11490, Taiwan; chengcc@mail.ndmctsgh.edu.tw (C.-C.C.); ljtmail@gmail.com (J.-T.L.); wslin545@ms27.hinet.net (W.-S.L.); dmscmsc@yahoo.com.tw (S.-M.C.); 7Department of Emergency Medicine, Tri-Service General Hospital, National Defense Medical Center, Taipei 11490, Taiwan; syjou.chen@gmail.com; 8Graduate Institute of Injury Prevention and Control, College of Public Health and Nutrition, Taipei Medical University, Taipei 11031, Taiwan; 9Division of Cardiology, Cheng Hsin General Hospital, Taipei 11220, Taiwan

**Keywords:** statins, new-onset diabetes mellitus, atherosclerotic cardiovascular disease

## Abstract

Statins constitute the mainstay treatment for atherosclerotic cardiovascular disease, which is associated with the risk of new-onset diabetes mellitus (NODM). However, the effects of individual statins on the risk of NODM remain unclear. We recruited 48,941 patients taking one of the three interested statins in a tertiary hospital between 2006 and 2018. Among them, 8337 non-diabetic patients taking moderate-intensity statins (2 mg/day pitavastatin, 10 mg/day atorvastatin, and 10 mg/day rosuvastatin) were included. The pitavastatin group had a higher probability of being NODM-free than the atorvastatin and rosuvastatin groups during the 4-year follow-up (log-rank test: *p* = 0.038). A subgroup analysis revealed that rosuvastatin had a significantly higher risk of NODM than pitavastatin among patients with coronary artery disease (CAD) (adjusted HR [aHR], 1.47, 95% confidence interval [CI], 1.05–2.05, *p* = 0.025), hypertension (aHR, 1.26, 95% CI, 1.00–1.59, *p* = 0.047), or chronic obstructive pulmonary disease (COPD) (aHR, 1.74, 95% CI, 1.02–2.94, *p* = 0.04). We concluded that compared with rosuvastatin, reduced diabetogenic effects of pitavastatin were observed among patients treated with moderate-intensity statin who had hypertension, COPD, or CAD. Additional studies are required to prove the effects of different statins on the risk of NODM.

## 1. Introduction

Statins, also known as 3-hydroxy-3-methylglutaryl-coenzyme A reductase inhibitors, constitute the mainstay pharmacologic treatment for dyslipidemia to prevent atherosclerotic cardiovascular disease (ASCVD) [1,2]. Current guidelines recommend targeting a lower low-density lipoprotein cholesterol (LDL-C) level for individuals with higher cardiovascular risk and intensifying statin treatment before adding ezetimibe to the treatment regimen [1,2]. Findings from a study of 26 randomized trials involving 170,000 patients suggested that lowering LDL-C by 1 mmol/L (approximately 38.7 mg/dL) reduced the risk of major vascular events by 22% and reduced deaths from coronary artery disease (CAD) by 20% [3]. Although statins are effective and safe [4], several studies have raised concerns regarding the risk of new-onset diabetes mellitus (NODM) after statin therapy [5,6,7,8]. Because the therapeutic target of LDL-C is lower in current guidelines than the previous ones, clinicians should be aware of the risk of NODM associated with statin therapy.

The effects of different statins on the risk of NODM are controversial. Previous studies have highlighted the class effects of statins on NODM development and glucose metabolism [9,10]. However, the J-PREDICT study including 1269 participants with impaired glucose tolerance (IGT) revealed a lower diabetes risk from IGT in the pitavastatin group than in the lifestyle modification group (HR, 0.82, 95% CI, 0.68–0.99) [11], which indicated the different effects of individual statins on the risk of NODM. A network meta-analysis of 27 clinical trials comparing the effects of different statins suggested a higher risk of NODM associated with high-intensity atorvastatin (odds ratios [OR], 1.34, 95% confidence interval [CI], 1.14–1.57) and rosuvastatin (OR, 1.17, 95% CI, 1.02–1.35) than with simvastatin, pravastatin, lovastatin, and pitavastatin [12]. Due to a small sample size and the short follow-up duration, these studies might have failed to prove the long-term diabetogenic effects of individual statins. Pitavastatin was proposed to have the highest risk of NODM compared with rosuvastatin, atorvastatin, pravastatin, and simvastatin in a retrospective cohort study, with a follow-up period of 5 years [8]. However, in a 19-year retrospective cohort study, pitavastatin was associated with a lower HR of NODM than atorvastatin, fluvastatin, pravastatin, rosuvastatin, and simvastatin, without statistical significance [13]. Such evidence indicates the controversial effects of individual statins on the risk of NODM.

In this study, we investigated the effects of commonly used moderate-intensity statins (pitavastatin, atorvastatin, and rosuvastatin) on the risk of NODM by utilizing a single-institute real-world database from 2006 to 2018, containing more than 48,000 patients. The predictors for NODM in the study were evaluated and specific subgroup analysis were conducted.

## 2. Material and Methods

### 2.1. Population

This retrospective cohort study was conducted using a clinical database of electronic medical records (EMRs) curated by the Tri-Service General Hospital, a tertiary hospital in Taiwan. From the database, a total of 48,941 patients treated with atorvastatin, rosuvastatin, or pitavastatin for the first time between January 2006 and July 2018 were included. This study was approved by the Institutional Review Board of Tri-Service General Hospital (IRB NO 1-108-05-193). Extracted data comprised information on gender; age; body mass index (BMI); daily dose and initiation date of statins; baseline comorbidities based on International Classification of Diseases, Ninth Revision and Tenth Revision (ICD-9 and ICD-10, respectively) codes; and baseline biochemistry data on the levels of serum creatinine, alanine aminotransferase (ALT), cholesterol, uric acid, fasting glucose, and glycated hemoglobin (HbA1c).

The prescription of either atorvastatin, rosuvastatin, or pitavastatin was based on the real-world practices and the guidelines of that time to reach nearly random with considerable patients included. We excluded patients exposed to more than one type of statin (atorvastatin, rosuvastatin, pitavastatin, simvastatin, fluvastatin, pravastatin, and lovastatin) during the follow-up period. Consecutive prescriptions were defined as continuous identical prescription for 2 months or more. Patients without consecutive prescriptions were excluded. Additionally, to focus on moderate-intensity statins, patients administered with daily dosages of statins other than 10 mg/day atorvastatin, 10 mg/day rosuvastatin, and 2 mg/day pitavastatin were excluded. Patients with ICD-9 or ICD-10 diagnosis codes related to diabetes mellitus (DM) and those who had either two consecutive fasting glucose levels of 126 mg/dL or higher or two consecutive HbA1c levels of 6.5% or higher before statin therapy were also excluded. To reduce potential biases in the real-world data, including drug switching, different dosing, and pre-existing diabetes, we set a selection process as shown in Figure 1. Finally, 8337 patients were included in the sample population.

### 2.2. Observational Variables

The endpoint of this study was the development of NODM, which was defined as a new diagnosis of type 2 DM with ICD-9 codes 250.x0 and 250.x2 or the ICD-10 code E11, two consecutive fasting glucose levels of 126 mg/dL or higher, or two consecutive HbA1c levels of 6.5% or higher during the follow-up period. The index date was the development of DM. Patients were followed up from the first day of consecutive prescriptions to DM development, last prescription date of statins, or 7 July 2018.

We collected information on gender, age, BMI, baseline biochemistry, and baseline comorbidities for risk evaluation and comparison. Baseline biochemistry was obtained within 90 days before and after enrollment. Baseline comorbidities were extracted using ICD-9 and ICD-10 codes, including CAD (ICD-9 codes 410–414 or ICD-10 codes I20–I25), hypertension (ICD-9 codes 401–405 or ICD-10 codes I10–I16), congestive heart failure (ICD-9 codes 428, 398.91, and 402.x1 or ICD-10 codes I50, I09.8, and I11.0), COPD (ICD-9 codes 490–492 and 494–496 or ICD-10 codes J40–J44), chronic kidney disease (CKD) (ICD-9 code 585 or ICD-10 codes N18), cancer (ICD-9 codes 140–165, 170–176, and 179–209 or ICD-10 code C), hemorrhagic stroke (ICD-9 codes 430–432 or ICD-10 codes I60–I62), and ischemic stroke (ICD-9 codes 433–438 or ICD-10 codes I63–I66).

### 2.3. Statistical Analysis

The characteristics are presented as means and standard deviations, numbers of patients, or percentages, as appropriate. We compared this information using either the chi-square test or analysis of variance, as appropriate. To determine potential risk factors for NODM, gender, age, baseline comorbidities, baseline biochemistry data, and statins exposure were analyzed using Cox proportional hazard regression models. Hazard ratios (HRs) were adjusted by gender, age, CAD, hypertension, COPD, CKD, cancer, ischemic stroke, hemorrhagic stroke, and congestive heart failure. Additional adjustment on HRs by baseline biochemistry were performed (Appendix A).

Because pitavastatin was launched a few years later than atorvastatin and rosuvastatin, adjustment was made for patients enrolled before and after the release of pitavastatin. Furthermore, we performed sensitivity tests to exclude patients enrolled before the year in which pitavastatin was launched (in 2013) and those who developed NODM shortly (30 or 90 days) after statin therapy (Appendix A).

Pre-existing diseases before statin therapy might have had different effects on statin-induced NODM, as reported in previous studies [8,14,15]. We stratified our patients based on baseline comorbidities and compared the diabetogenic effect of the three statins in each subgroup. Cox proportional hazard regression models and the Kaplan–Meier method with the log-rank test were used for comparing the difference in the risk of NODM in different statin groups. The data were analyzed using R software, version 3.4.4 (R Foundation for Statistical Computing, Vienna, Austria). Two-tailed *p* values < 0.05 indicated statistical significance in all analyses.

## 3. Results

We identified 1312 (15.7%) patients as pitavastatin users, 3034 (36.4%) patients as atorvastatin users, and 3991 (47.9%) patients as rosuvastatin users. Table 1 demonstrates the characteristics of the patients in each group. The majority of our population was male (54.1%). The mean follow-up time was 587.22 ± 707.63 days, and the mean age was 59.77 ± 13.42 years old. The cumulative incidence of NODM in the study period was 12.7% in the pitavastatin group, 18.3% in the atorvastatin group, and 21.6% in the rosuvastatin group (*p* < 0.001). The pitavastatin group had a higher proportion of patients with CAD (42%) and hypertension (60.7%) compared with the atorvastatin and rosuvastatin groups (*p* < 0.001 for CAD and hypertension). Baseline levels of LDL, cholesterol, and triglyceride were higher in the rosuvastatin group than in the other two groups (*p* < 0.001 for LDL, cholesterol, and triglyceride). After treatment, the LDL levels reduced 23%, 16%, and 22% in pitavastatin, atorvastatin, and rosuvastatin group, respectively (Appendix A). Fasting glucose and HbA1c levels showed no significant difference between the three groups at baseline (*p* = 0.583 and *p* = 0.872, respectively).

We analyzed baseline characteristic risk factors for NODM through a univariable Cox regression model (Table 2). After adjustment, the development of NODM was more common among male patients (adjusted HR [aHR], 1.17, 95% CI, 1.05–1.29, *p* = 0.003) and older patients (aHR, 1.02, 95% CI, 1.01–1.02, *p* < 0.001). Analysis of comorbidities associated with NODM revealed protective effects for CAD (aHR, 0.63, 95% CI, 0.55–0.72, *p* < 0.001), hypertension (aHR, 0.78, 95% CI, 0.70–0.87, *p* < 0.001), and COPD (aHR, 0.71, 95% CI, 0.59–0.86, *p* < 0.001). However, CKD increased the risk of NODM (aHR, 1.56, 95% CI, 1.20–2.02, *p* = 0.001). In baseline biochemistry data, HbA1c (aHR, 1.64, 95% CI, 1.57–1.73, *p* < 0.001) and creatinine (aHR, 1.14, 95% CI, 1.07–1.23, *p* < 0.001) showed a positive association with the development of NODM, whereas LDL and cholesterol had an inverted association with the development of NODM (LDL: aHR, 0.97, 95% CI, 0.96–0.99, *p* = 0.006, cholesterol: aHR, 0.97, 95% CI, 0.9–0.99, *p* = 0.001).

Associations between prescribed statins and the risk of NODM were compared using the Cox regression model and Kaplan–Meier curve. Compared with the pitavastatin group, the atorvastatin and rosuvastatin groups showed a higher risk of NODM (HR, 1.21, 95% CI, 1.02–1.44, *p* = 0.032 and HR, 1.24, 95% CI, 1.05–1.47, *p* = 0.011, respectively, Table 3). Moreover, the group of combined atorvastatin and rosuvastatin was associated with a higher risk of NODM than that of pitavastatin (HR, 1.23, 95% CI, 1.05–1.45, *p* = 0.012). The HRs in the three statin groups were not statistically significant after adjusting for baseline characteristics and comorbidities (*p* = 0.23). To evaluate the trend of NODM after exposure to different statins, the NODM-free probabilities among the statin groups during the 4-year follow-up period were evaluated (Figure 2). The NODM-free probabilities were significantly higher in the pitavastatin group than in the atorvastatin and rosuvastatin groups (log-rank test: *p* = 0.038) and combined groups (log-rank test: *p* = 0.043).

Because gender and certain comorbidities influenced the development of NODM in our study, we compared the HR in these three statin groups, which were stratified by gender and specific diseases (Table 4). Compared with pitavastatin, rosuvastatin was associated with a significantly higher risk of NODM in patients with CAD (aHR, 1.47, 95% CI, 1.05–2.05, *p* = 0.025), hypertension (aHR, 1.26, 95% CI, 1.00–1.59, *p* = 0.047), or COPD (aHR, 1.74, 95% CI, 1.02–2.94, *p* = 0.04). Compared with pitavastatin, rosuvastatin was associated with NODM development in patients with CKD and heart failure and in female patients, without statistical significance (*p* = 0.063, *p* = 0.064, and *p* = 0.066, respectively). Due to some degrees of missing values, LDL, cholesterol, and triglyceride were adjusted additionally, which revealed a similar trend with the previous results (Appendix A). Finally, sensitivity tests were conducted, which showed a trend of lower NODM incidence in the pitavastatin group than in the atorvastatin or rosuvastatin groups (Appendix A).

## 4. Discussion

In our study, we observed a trend of lower risk of NODM in the pitavastatin group than in the atorvastatin and rosuvastatin groups. Moreover, patients with CAD, hypertension, or COPD demonstrated a significantly reduced risk of NODM, whereas patients with CKD demonstrated a significantly increased risk of NODM. After adjusting for all confounding factors, serum creatinine levels significantly predicted the risk of NODM in our study. Importantly, we first demonstrated that patients with CAD, hypertension, or COPD in the pitavastatin group had a significantly lower risk of NODM than those in the rosuvastatin group.

The definite molecular mechanisms of statin-induced NODM were unclear. Previous studies have described two possible mechanisms: altering pancreatic β-cell function and decreasing systemic insulin sensitivity [16]. Statins involve pathways of pancreatic β-cell function and insulin secretion, such as inhibition of voltage-gated calcium channels [17], inflammation of the pancreatic islet [18], dysfunction of mitochondria [19], and alteration of coenzyme Q10 levels [20]. Moreover, several mechanisms explain the effects of statins on reducing insulin sensitivity, such as reducing HMG-CoA reductase activity [21], suppressing glucose transporter 4 (GLUT-4) [22], decreasing the release of adiponectin [23], and increasing fat and caloric intake [24]. Although structural differences based on lipophilic or hydrophilic features have been proposed to be the possible mechanisms of stain-induced NODM [25], clinical studies have demonstrated no apparent difference in the diabetogenic effects between lipophilic and hydrophilic statins [8,26]. Arnaboldi L. et al. summarized the effects of different statins on the plasma adiponectin concentration; they concluded that pitavastatin seemed to be the most effective in increasing plasma adiponectin compared with simvastatin, pravastatin, fluvastatin, atorvastatin, and rosuvastatin, which is potentially conducive to glucose metabolism [27]. Such evidence elucidates the protective role of pitavastatin on the risk of NODM in our study.

High-intensity statin therapy was found to pose a higher risk of NODM compared with moderate-intensity therapy (OR, 1.12, 95% CI, 1.04–1.22) [28]. Interestingly, in the Asian population, a lower dose of statins may reduce ASCVD similar to the benefit of higher doses in the Western population [29]. Such evidence points out the vital role of moderate intensity statin treatment in Asian patients with ASCVD.

Previous studies have provided controversial findings regarding whether pitavastatin is associated with lower risk of NODM than atorvastatin and rosuvastatin [8,9,13,15,30,31]. Although Vallejo-Vaz et al. proposed a neutral effect of pitavastatin on the risk of NODM through a meta-analysis of randomized control trials, the included trials might have included relatively small sample sizes and short follow-up periods, which might have limited the diabetogenic effect of statins [31]. In a cohort study, Cho et al. proposed a higher risk of NODM in the pitavastatin group than in the atorvastatin and rosuvastatin groups, without statistical significance, during the 5-year follow-up period in Korea [8]. Notably, Kim et al. studied the Korean population and suggested the lowest rate of progression from non-diabetes to prediabetes in the pitavastatin group during the 4-year follow-up period [30]. The baseline characteristics and comorbidities of the included population in each study might have led to the discrepancy in the results [8,9,13,30]. In our study with a 4-year follow-up period, we enrolled 8337 patients. The DM-free survival curve between pitavastatin group and the combined atorvastatin and rosuvastatin group spreads apart after 1.5 years of follow-up, indicating the long-term effects of statins on DM development (Figure 2b). Furthermore, studies on patients with acute myocardial infarction have revealed a significantly lower risk of NODM with pitavastatin compared with atorvastatin and rosuvastatin, which is consistent with our findings [15]. Along with our results, such evidence highlights the protective effects of pitavastatin on the risk of NODM.

Previous studies have rarely discussed the influence of underlying comorbidities on the risk of statin-induced NODM. Our study proposed the significantly beneficial effects of pitavastatin on NODM in patients with CAD, hypertension, or COPD compared with rosuvastatin. The mechanisms linking the protective effects of pitavastatin on the risk of NODM in patients with CAD, hypertension, or COPD warrant further investigation. Systemic inflammation, which is common in patients with ASCVD and COPD, was found to be a crucial factor in insulin resistance and DM development [32]. Statins exert potent pleiotropic effects on inflammation through the suppression of inflammatory cell activation and inhibition of cytokine production [33,34]. Interestingly, pitavastatin exerts possibly stronger anti-inflammatory and immunomodulatory effects than atorvastatin and rosuvastatin through the suppression of extracellular signal-regulated kinase (ERK) and p38 mitogen-activated protein kinases (MAPK) signaling and the reduction of activating protein-1 (AP-1) pathways in human T cells [35]. Furthermore, a 12-month follow-up randomized control trial revealed that inflammatory chemokines and cytokines which are related to DM development [36], such as monocyte chemoattractant protein-1 (MCP-1) and tumor necrosis factor alpha (TNF-α), were significantly reduced in the pitavastatin group than in the atorvastatin group [37]. These findings may partly explain the risk of NODM between the three stain groups with certain diseases.

Lifestyle-related chronic illnesses such as CKD, CAD, hypertension, and COPD are risk factors for incident diabetes due to common coexistence of obesity, smoking, and metabolic dysfunction [38,39,40]. Our study indicated that serum creatinine levels and CKD were independent predictors for NODM. However, hypertension, CAD, and COPD were protective factors for NODM in our population after excluding overt DM. Although many confounding factors might have existed in the observational studies, we proposed that statin users with these chronic inflammatory diseases but without overt diabetes might have lower NODM probability than those without these diseases. Additional large-scale studies are required to verify the effects of these comorbidities on the risk of NODM during statin treatment and to explore the underlying mechanisms.

This study had some limitations. First, this study was retrospective in nature although we made adjustments for gender, age, and baseline comorbidities to reduce the confounding factors in the statin groups. Second, this study was conducted at a single institute in Taiwan. Hence, the results should be applied to other populations with caution. Similar evidence for different ethnic groups is required. Third, some missing laboratory data and body mass index (BMI) values in our database limited the analysis of those variables and their statistical power. Additionally, the effects of other potentially pro-diabetogenic drugs, including beta-blockers, diuretics, steroids, and antidepressant agents, on the risk of NODM were not evaluated in the current study [41]. Finally, it was challenging to analyze the causes of loss to follow up or a switching to other statins in this EMR-based retrospective study. However, we set up a strict study protocol and performed sensitivity tests (Appendix A) to reduce the bias in observational studies as much as possible. Even with these limitations, our studies provide practical clues to treat hyperlipidemia in patients with ASCVD.

In conclusion, among patients treated with moderate-intensity statins, a nonsignificant trend of lower risk of NODM was associated with pitavastatin compared with atorvastatin and rosuvastatin based on a real-world clinical database. Patients with hypertension, COPD, or CAD might benefit from pitavastatin in terms of reduced NODM development, with statistical significance. On the basis of our results, clinicians can select an appropriate statin according to patients’ comorbidities and baseline characteristics. Additional large-scale randomized trials are required to verify the relationship between different statins and NODM in patients with certain diseases.

## Figures and Tables

**Figure 1 biomedicines-08-00499-f001:**
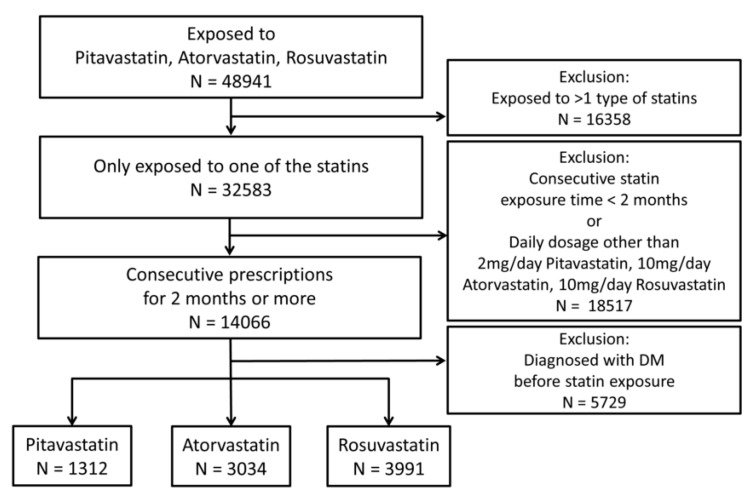
Flowchart of study population enrollment based on the inclusion and exclusion criteria. DM: diabetes mellitus; N: number of patients.

**Figure 2 biomedicines-08-00499-f002:**
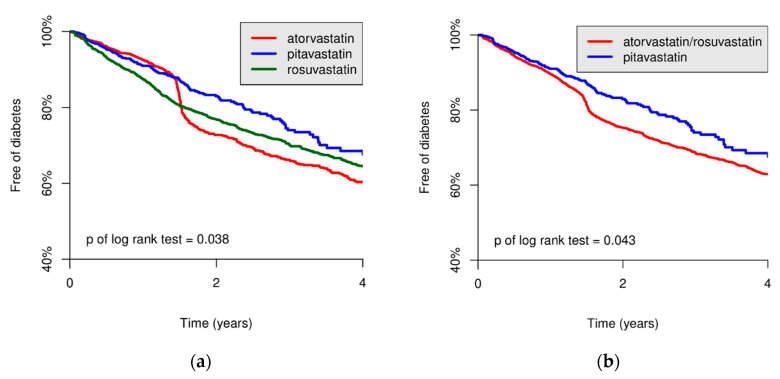
Kaplan–Meier plot of three statin groups for new-onset diabetes mellitus (NODM)-free probabilities curve. The NODM-free probabilities were significantly higher in the pitavastatin group than in the atorvastatin and rosuvastatin groups (log-rank test: *p* = 0.038) (**a**) and the group of combined atorvastatin and rosuvastatin (log-rank test: *p* = 0.043) (**b**).

**Table 1 biomedicines-08-00499-t001:** Baseline characteristics of three statin groups.

Variables	Pitavastatin (N = 1312, 15.7%)	Atorvastatin (N = 3034, 36.4%)	Rosuvastatin (N = 3991, 47.9%)	*p*-Value
Follow up (days)	468.31 ± 446.18	558.30 ± 666.85	648.30 ± 796.06	<0.001
NODM	167 (12.7%)	555 (18.3%)	864 (21.6%)	<0.001
Gender (male)	728 (55.5%)	1546 (51.0%)	2236 (56.0%)	<0.001
Age	60.42 ± 12.44	61.53 ± 13.54	58.21 ± 13.45	<0.001
BMI	25.71 ± 4.18	24.75 ± 4.08	25.11 ± 4.38	0.017
**Comorbidities**				
CAD	551 (42.0%)	798 (26.3%)	931 (23.3%)	<0.001
Hypertension	796 (60.7%)	1416 (46.7%)	1864 (46.7%)	<0.001
COPD	195 (14.9%)	363 (12.0%)	419 (10.5%)	<0.001
CKD	29 (2.2%)	154 (5.1%)	90 (2.3%)	<0.001
Cancer	82 (6.2%)	313 (10.3%)	266 (6.7%)	<0.001
Ischemic stroke	160 (12.2%)	508 (16.7%)	736 (18.4%)	<0.001
Hemorrhagic stroke	24 (1.8%)	76 (2.5%)	71 (1.8%)	0.086
Heart failure	100 (7.6%)	213 (7.0%)	256 (6.4%)	0.279
**Biochemistry**				
LDL (mg/dL)	131.52 ± 28.51	122.94 ± 37.57	141.30 ± 45.86	<0.001
TC (mg/dL)	202.91 ± 35.03	198.53 ± 46.67	218.96 ± 54.14	<0.001
TG (mg/dL)	138.95 ± 74.01	138.80 ± 84.14	164.14 ± 133.90	<0.001
Creatinine (mg/dL)	0.89 ± 0.27	1.03 ± 0.98	0.94 ± 0.64	<0.001
ALT (U/L)	23.59 ± 16.43	23.64 ± 20.52	25.78 ± 19.71	<0.001
Uric Acid (mg/dL)	6.07 ± 1.56	6.05 ± 1.66	6.13 ± 1.79	0.376
FG (mg/dL)	99.94 ± 15.74	99.27 ± 17.31	99.57 ± 17.98	0.581
HbA1c (%)	6.05 ± 0.98	6.08 ± 1.17	6.09 ± 1.23	0.872

ALT: alanine aminotransferase; BMI: body mass index; CAD: coronary artery disease; CKD: chronic kidney disease; COPD: chronic obstructive pulmonary disease; FG: fasting glucose; HbA1c: glycated hemoglobin; LDL: low-density lipoprotein; N: number of patients; NODM: new-onset diabetes mellitus; TC: total cholesterol; TG: triglyceride.

**Table 2 biomedicines-08-00499-t002:** Hazard ratio (HR) of baseline characteristics in new-onset diabetes mellitus (NODM).

Variables	Crude-HR (95% CI)	*p*-Value	Adj-HR (95% CI) #	*p*-Value
Gender (male)	1.00 (0.91–1.10)	0.987	1.17 (1.05–1.29)	0.003
Age	1.01 (1.01–1.02)	<0.001	1.02 (1.01–1.02)	<0.001
**Comorbidities**				
CAD	0.64 (0.57–0.73)	<0.001	0.63 (0.55–0.72)	<0.001
Hypertension	0.81 (0.73–0.89)	<0.001	0.78 (0.70–0.87)	<0.001
COPD	0.73 (0.61–0.88)	0.001	0.71 (0.59–0.86)	<0.001
CKD	1.58 (1.22–2.04)	0.001	1.56 (1.20–2.02)	0.001
cancer	0.93 (0.76–1.12)	0.433	0.83 (0.69–1.01)	0.067
Ischemic stroke	1.16 (1.02–1.32)	0.028	1.00 (0.87–1.16)	0.964
Hemorrhagic stroke	1.06 (0.73–1.54)	0.761	0.99 (0.68–1.45)	0.969
Heart failure	1.16 (0.96–1.40)	0.113	1.17 (0.96–1.42)	0.121
**Biochemistry**				
LDL (per 10 mg/dL)	0.97 (0.95–0.99)	<0.001	0.97 (0.96–0.99)	0.006
TC (per 10 mg/dL)	0.97 (0.95–0.98)	<0.001	0.97 (0.96–0.99)	0.001
TG (per 10 mg/dL)	1.01 (1.01–1.02)	<0.001	1.02 (1.01–1.02)	<0.001
Cr (per 1 mg/dL)	1.18 (1.12–1.25)	<0.001	1.14 (1.07–1.23)	<0.001
ALT (per 1 U/L)	1.01 (1.00–1.01)	<0.001	1.01 (1.01–1.01)	<0.001
UA (per 1 mg/dL)	1.06 (1.02–1.11)	0.006	1.05 (1.00–1.09)	0.043
FG (per 1 mg/dL)	1.03 (1.03–1.03)	<0.001	1.03 (1.03–1.03)	<0.001
HbA1c (per 1%)	1.63 (1.55–1.70)	<0.001	1.64 (1.57–1.73)	<0.001

#: All results of Adj-HR were adjusted by years, gender, age, CAD, hypertension, COPD, CKD, cancer, ischemic stroke, hemorrhagic stroke, heart failure. Adj-HR: adjusted hazard ratio; ALT: alanine aminotransferase; CAD: coronary artery disease; CI: confidence interval; CKD: chronic kidney disease; COPD: chronic obstructive pulmonary disease; Cr: Creatinine; FG: fasting glucose; HbA1c: glycated hemoglobin; HR: hazard ratio; LDL: low-density lipoprotein; NODM: new-onset diabetes mellitus; TC: total cholesterol; TG: triglyceride; UA: uric acid; years: years before and after the release of pitavastatin. In LDL, TC, and TG, per 10 mg/dL indicates that every additional 10 mg/dL at baseline reduces 3% (HR, 0.97), and increases 1% (HR, 1.01) of the risk of NODM; In Cr, UA, and FG, per 1 mg/dL indicates that every additional 1 mg/dL at baseline increases 18% (HR, 1.18), 6% (HR, 1.06), and 3% (HR, 1.03) of the risk of NODM; In ALT, per 1 U/L indicates that every additional 1 U/L at baseline increases 1% (HR, 1.01) of the risk of NODM; in HbA1c, per 1% indicates that every additional 1% at baseline increases 63% (HR, 1.63) of the risk of NODM.

**Table 3 biomedicines-08-00499-t003:** HR of three statin groups in NODM.

Independent Variables	Crude-HR (95% CI)	*p*-Value	Adj-HR (95% CI) #	*p*-Value
Comparison 1		0.038		0.230
Pitavastatin	1.00		1.00	
Atorvastatin	1.21 (1.02–1.44)	0.032	1.04 (0.87–1.25)	0.677
Rosuvastatin	1.24 (1.05–1.47)	0.011	1.13 (0.94–1.35)	0.196
Comparison 2				
Pitavastatin	1.00		1.00	
Atorvastatin/Rosuvastatin	1.23 (1.05–1.45)	0.012	1.09 (0.91–1.29)	0.356

#: All results of Adj-HR were adjusted by years, gender, age, coronary artery disease, hypertension, chronic obstructive pulmonary disease, chronic kidney disease, cancer, ischemic stroke, hemorrhagic stroke, heart failure. Adj-HR: adjusted hazard ratio; CI: confidence interval; HR: hazard ratio; NODM: new-onset diabetes mellitus; years: years before and after the release of pitavastatin.

**Table 4 biomedicines-08-00499-t004:** HR of three statin groups in NODM stratified with baseline characteristics.

Stratified Variables	Drugs	Crude-HR (95% CI)	*p*-Value	Adj-HR (95% CI) #	*p*-Value
Gender					
Female	Pitavastatin	1.00		1.00	
(N = 3827)	Atorvastatin	1.25 (0.96–1.64)	0.103	1.07 (0.81–1.43)	0.627
	Rosuvastatin	1.43 (1.10–1.86)	0.007	1.31 (0.99–1.73)	0.063
Male	Pitavastatin	1.00		1.00	
(N = 4510)	Atorvastatin	1.19 (0.95–1.50)	0.131	1.03 (0.81–1.31)	0.792
	Rosuvastatin	1.12 (0.90–1.39)	0.317	1.02 (0.80–1.29)	0.899
Comorbidities					
No CAD	Pitavastatin	1.00		1.00	
(N = 6057)	Atorvastatin	1.09 (0.88–1.34)	0.431	0.93 (0.75–1.16)	0.535
	Rosuvastatin	1.12 (0.91–1.37)	0.279	1.01 (0.81–1.25)	0.926
CAD	Pitavastatin	1.00		1.00	
(N = 2280)	Atorvastatin	1.17 (0.84–1.62)	0.365	1.25 (0.89–1.74)	0.202
	Rosuvastatin	1.16 (0.85–1.59)	0.357	1.47 (1.05–2.05)	0.025
No HTN	Pitavastatin	1.00		1.00	
(N = 4261)	Atorvastatin	1.37 (1.03–1.82)	0.031	0.87 (0.64–1.19)	0.379
	Rosuvastatin	1.44 (1.09–1.90)	0.011	0.99 (0.73–1.34)	0.934
HTN	Pitavastatin	1.00		1.00	
(N = 4076)	Atorvastatin	1.03 (0.82–1.30)	0.778	1.13 (0.90–1.43)	0.295
	Rosuvastatin	1.05 (0.84–1.30)	0.688	1.26 (1.00–1.59)	0.047
No COPD	Pitavastatin	1.00		1.00	
(N = 7360)	Atorvastatin	1.23 (1.02–1.48)	0.028	1.04 (0.86–1.27)	0.691
	Rosuvastatin	1.22 (1.02–1.46)	0.026	1.09 (0.90–1.32)	0.381
COPD	Pitavastatin	1.00		1.00	
(N = 977)	Atorvastatin	0.90 (0.52–1.55)	0.698	0.99 (0.57–1.72)	0.965
	Rosuvastatin	1.31 (0.79–2.16)	0.292	1.74 (1.02–2.94)	0.040
No CKD	Pitavastatin	1.00		1.00	
(N = 8064)	Atorvastatin	1.17 (0.98–1.40)	0.077	1.00 (0.83–1.21)	0.999
	Rosuvastatin	1.22 (1.03–1.44)	0.024	1.08 (0.90–1.30)	0.385
CKD	Pitavastatin	1.00		1.00	
(N = 273)	Atorvastatin	2.54 (0.78–8.32)	0.123	2.67 (0.79–8.99)	0.114
	Rosuvastatin	3.08 (0.92–10.3)	0.068	3.33 (0.93–11.9)	0.064
No cancer	Pitavastatin	1.00		1.00	
(N = 7676)	Atorvastatin	1.28 (1.07–1.54)	0.007	1.06 (0.88–1.29)	0.549
	Rosuvastatin	1.25 (1.05–1.49)	0.012	1.09 (0.90–1.32)	0.361
Cancer	Pitavastatin	1.00		1.00	
(N = 661)	Atorvastatin	0.65 (0.35–1.21)	0.176	0.92 (0.48–1.75)	0.797
	Rosuvastatin	1.21 (0.66–2.22)	0.534	1.62 (0.85–3.08)	0.145
No IS	Pitavastatin	1.00		1.00	
(N = 6933)	Atorvastatin	1.28 (1.06–1.54)	0.011	1.06 (0.86–1.29)	0.601
	Rosuvastatin	1.31 (1.09–1.58)	0.004	1.16 (0.95–1.42)	0.149
IS	Pitavastatin	1.00		1.00	
(N = 1404)	Atorvastatin	0.82 (0.53–1.28)	0.384	0.89 (0.57–1.40)	0.617
	Rosuvastatin	0.87 (0.57–1.33)	0.520	1.01 (0.65–1.56)	0.972
No HS	Pitavastatin	1.00		1.00	
(N = 8166)	Atorvastatin	1.22 (1.03–1.46)	0.024	1.05 (0.87–1.27)	0.582
	Rosuvastatin	1.26 (1.07–1.50)	0.007	1.15 (0.96–1.38)	0.142
HS	Pitavastatin	1.00		1.00	
(N = 171)	Atorvastatin	0.56 (0.18–1.79)	0.328	0.43 (0.12–1.49)	0.183
	Rosuvastatin	0.50 (0.16–1.63)	0.251	0.38 (0.10–1.42)	0.151
No HF	Pitavastatin	1.00		1.00	
(N = 7768)	Atorvastatin	1.20 (1.00–1.44)	0.045	1.00 (0.82–1.21)	0.960
	Rosuvastatin	1.24 (1.04–1.48)	0.015	1.08 (0.90–1.31)	0.407
HF	Pitavastatin	1.00		1.00	
(N = 569)	Atorvastatin	1.29 (0.70–2.36)	0.411	1.54 (0.83–2.87)	0.173
	Rosuvastatin	1.30 (0.72–2.34)	0.384	1.82 (0.96–3.43)	0.066

#: All results of Adj-HR were adjusted by years, gender, age, CAD, hypertension, COPD, CKD, cancer, ischemic stroke, hemorrhagic stroke, heart failure. Adj-HR: adjusted hazard ratio; CAD: coronary artery disease; CI: confidence interval; CKD: chronic kidney disease; COPD: chronic obstructive pulmonary disease; HF: heart failure; HR: hazard ratio; HS: hemorrhagic stroke; HTN: hypertension; IS: ischemic stroke; N: number of patients; NODM: new-onset diabetes mellitus; years: years before and after the release of pitavastatin.

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
