# Peer review of "Effects of Pitavastatin, Atorvastatin, and Rosuvastatin on the Risk of New-Onset Diabetes Mellitus: A Single-Center Cohort Study"

_biomedicines, 2020, doi:10.3390/biomedicines8110499_

Round 1
Reviewer 1 Report
Wei-Ting Liu et al investigated 8337 non-diabetic patients at the onset of the survey taking the CES inhibitors pitavastatin , atorvastatin or rosuvastatin and evaluated the incidence of new-onset diabetes mellitus (NODM) after a 4-year follow-up. The overall conclusion was “that compared with rosuvastatin, reduced diabetogenic effects of pitavastatin were observed among patients treated with moderate-intensity statin who had hypertension, COPD, or CAD”.
GENERAL COMMENTS
The question as to what extent different statins may be diabetogenic is highly controversial and covered in numerous previous studies, most of them correctly cited in this report. The question therefore is, does this report add significantly to our knowledge in this area and does this justify publication in the journal Biomedicines.
I am in the opinion that the answer is YES . There are however several points that need consideration by the authors.
Overall the manuscript is very well written and conscious presented. The Limitations of the study are well taken and comprehensively presented.
SPECIFIC COMMENTS
- From the 48941 patients screened in the database only about 17% were left to be finally included. The question is whether the selection process was all to rigorous.
- Years of follow-up and number of patients included in the calculations: On page 8 the authors write: “ In our study with a 12-year follow-up period, we enrolled 48,941 patients” . This is misleading since the calculations are based on a 4 year follow-up and in addition only 8337 patients were included in the calculations. This should be corrected accordingly.
- It is well known that individual statins display a different efficacy. The question therefore is whether the three groups treated with individual statins had comparable LDL-C reductions within the 4 your follow-up. Are there any data available on this and if so they should be included.
- 5 Tab.2 : The terms in parenthesis (per 10 mg/dl, per 1 mg/dl …. ) are not appropriately explained in the legend. Please clarify!
- 9: The sentence “These findings partly explain the risk of NODM between the three stain groups with certain diseases” should be corrected to :These findings MAY partly explain the risk of NODM between the three stain groups with certain diseases..
Author Response
Please see the attachment.
To avoid the format differences between Word versions, please check the PDF file of the revised manuscript for comparison.
Thank you very much.

Reviewer 2 Report
I’ve read with attention the paper of Liu et al. that is potentially of interest. The background and aim of the study have been clearly defined. The methodology applied is overall correct, the results are reliable. I think that only discussion could be improved. From one side, the authors should stress the concept that in previous studies high-intensity (and not moderate intensity) statin treatment is associated with incident diabetes. On the other side, the data could be read also in view of the use of diuretics and beta-blockers in the different statin treated groups (as well as other potentially pro-diabetogenic drugs...)
Author Response

(The authors gave the same response as above.)

Round 2
Reviewer 1 Report
No more comments from my side